# Detecting Sentiment toward Emerging Infectious Diseases on Social Media: A Validity Evaluation of Dictionary-Based Sentiment Analysis

**DOI:** 10.3390/ijerph19116759

**Published:** 2022-06-01

**Authors:** Sanguk Lee, Siyuan Ma, Jingbo Meng, Jie Zhuang, Tai-Quan Peng

**Affiliations:** 1Department of Communication, Michigan State University, East Lansing, MI 48824, USA; lswook555@gmail.com (S.L.); masiyua1@msu.edu (S.M.); jingbome@msu.edu (J.M.); 2Bob Schieffer College of Communication, Texas Christian University, Fort Worth, TX 76129, USA; jie.zhuang@tcu.edu

**Keywords:** infectious diseases, sentiment analysis, validity, LIWC, SentiWordNet, ANEW

## Abstract

Despite the popularity and efficiency of dictionary-based sentiment analysis (DSA) for public health research, limited empirical evidence has been produced about the validity of DSA and potential harms to the validity of DSA. A random sample of a second-hand Ebola tweet dataset was used to evaluate the validity of DSA compared to the manual coding approach and examine the influences of textual features on the validity of DSA. The results revealed substantial inconsistency between DSA and the manual coding approach. The presence of certain textual features such as negation can partially account for the inconsistency between DSA and manual coding. The findings imply that scholars should be careful and critical about findings in disease-related public health research that use DSA. Certain textual features should be more carefully addressed in DSA.

## 1. Introduction

Emotions play a significant role in people’s decision making when facing health crises. Despite the cumulative evidence of the influence of emotions on decision making and judgments produced from tightly controlled lab experiments or survey studies [1] little is known about how emotions are expressed on social media when health crises strike, nor are we extremely confident that the findings yielded by computational methods are valid.

Computational sentiment analysis complements manual content analysis by automatically detecting sentiments from large-scale user-initiated expressions [2]. One of the automatic sentiment analyses widely used for public health research is the off-the-shelf dictionary-based sentiment analysis (DSA, hereafter), which estimates sentiments by simply retrieving sentiment scores of words from a pre-defined sentiment dictionary [3]. Although advanced sentiment analyses such as neural network-based sentiment analyzers are introduced, DSA as an unsupervised approach is still widely utilized for sentiment detection of health documents due to its cost effectiveness, intuitiveness, and ease of use.

Reviewing journal articles and conference proceedings indexed in Web of Science, written in English, and published from 1 January 2000 to 28 June 2018 (see Appendix A for detailed information about the review search method), the authors found that DSA (*n* = 85) is more commonly adopted than supervised machine learning (*n* = 59) in sentiment detection in public health research. It is identified that domain-independent DSA such as Linguistic Inquiry and Word Count (LIWC, hereafter) (*n* = 16), SentiWordNet (SWN, hereafter) (*n* = 14), and Affective Norms for English Words (ANEW, hereafter) (*n* = 6) are the most popular DSAs used in public health research. The application of DSA in public health research covers a wide range of topics such as public sentiment responses to epidemic diseases [4], flu [5], and health-related products [6] as well as expressed sentiments in an online support community [7].

Despite the popularity of DSA across diverse topics of public health research, few studies in the public health context assess the validity of DSA in comparison to the manual coding approach, which is a well-established sentiment classification method. Some evaluation studies applying DSA to public health topics questioned the validity of DSA, particularly domain-independent applications, by showing their low-performance scores [8,9]. In other domains such as economics, a previous study explicitly concluded that DSA is invalid compared to the manual coding approach [10].

Past studies have suggested that a potential threat to DSA’s validity lies in its failure to account for unrecognizable textual features in human expressions [11]. These textual features can be broadly categorized into the semantic level and the word-level (see Table 1 for the summary of these textual features). Textual features at the semantic level include embedded hashtags, irrealis, sarcasm, negations, intensifiers, and diminishers. Word-level textual features include unconfirmed typos, lengthened words, irregularly capitalized words, abbreviations, and acronyms. Although past literature has acknowledged the lack of consideration of these potential textual features as a limitation of DSA that may impact the sentiment of texts, no empirical evaluation has been conducted to systematically examine if and to what extent these textual features will threaten the validity of DSA in detecting sentiment from human expressions in the context of health issues.

Uninformed use of sentiment analysis can have negative consequences on public health research, as it can provide inaccurate, inconsistent, or misleading evidence for policymakers and health professionals in their decision making. Therefore, it is practically and methodologically essential to assess the validity of DSA in detecting human sentiment from human expressions. Contextualized in online discussions of Ebola on social media, firstly, the current study evaluates the validity of DSA widely used in public health research by comparing them to the manual coding approach. Secondly, the current study aims to uncover textual features of texts that contribute to inconsistencies between DSA and the manual coding approach if any inconsistencies exist.

## 2. Methods

### 2.1. Data

The data of the study are a second-hand dataset obtained from a previous study on the spatial and temporal diffusion of Ebola tweets [20]. Specifically, the original dataset contains more than 17 million geo-tagged English tweets posted by more than 3 million users around the world from 20 July 2014 to 25 December 2014. The original dataset was collected via Twitter API by searching Ebola-related keywords.

The current study drew a small random sample from the original dataset. Two human coders were instructed to code the major sentiment embedded in the sampled tweets exclusively either positive, neutral, fear, anger, or sadness. In particular, a neutral sentiment tweet was explicitly defined in this study as a tweet without any emotional expressions. The coders were also instructed to evaluate whether a retweet included personal comments. Retweets without any personal comments were excluded in the analysis to focus on human expressed sentiments on the health subject. The inter-coder reliability for each coding category was computed with Krippendorff’s alpha [21]. The inter-coder reliabilities were 0.87 for positive sentiment, 0.92 for neutral sentiment, 0.93 for fear, 0.81 for anger, and 0.90 for sadness. To make the manual coding results comparable with that of DSA evaluated in the current study, fear, anger, or sadness tweets were relabeled as negative tweets. Someone may be concerned about relabeling fear as a negative sentiment, given its positive role in the public health context (e.g., fear activates cautions against risks). However, given that the study focuses on expressed emotion itself rather than the role of emotion, relabeling fear as negative sentiment is relevant. Moreover, previous studies suggest fear as a discrete emotion that belongs to the negative sentiment [22]. In total, 7799 tweets, including 1601 negative, 5729 neutral, and 469 positive sentiments, were retained as the benchmark dataset.

### 2.2. Target DSA

From our review above, LIWC, ANEW, and SWN were identified as popular DSA for public health research. Although not widely used in public health research, there are two of SWN’s extensions that can deal with word sense disambiguation, which is a technique that resolves the ambiguity of a word with multiple meanings [23]. The two applications use the original Lesk algorithm [24] (orgSWN, hereafter) and an adapted version of the original Lesk algorithm [25,26] (adSWN, hereafter), respectively, for word sense disambiguation. The current study focuses on evaluating these five DSA—LIWC, ANEW, SWN, orgSWN, and adSWN.

For LIWC, we used the official commercial version of LIWC 2015. For the other four applications, we employed Python packages or adapted Python code from the GitHub community. Text-preprocessing was conducted with the NLTK Python package to remove stopwords, convert all characters to lowercase, and lemmatize words. Then, for each tweet, both the positive and negative scores were estimated from each DSA.

### 2.3. Sentiment Classification of DSA

Each of the five applications estimate both positive and negative scores for each tweet. A common practice to classify a text into positive, negative, or neutral sentiment includes subtraction (when both scores have positive values) or summation (when positive scores have positive values, and negative scores have negative values) between positive and negative scores. This approach, however, blends the conceptualization of neutral sentiment by classifying a text as neutral either when there is no emotional expression at all or when positive scores are equal to negative scores. The current study explicitly defines the former case as the neutral sentiment, which is consistent with the definition used in the manual coding approach, and the latter case as the mixed sentiment. This conceptual clarification can reduce any potential contaminants that stem from obscure conceptualization of sentiments. Based on the conceptual definition above, the current study developed a classification rule to classify a tweet into positive, negative, neutral, or mixed sentiment (see Appendix A for detail). Tweets classified as mixed sentiment were excluded, as such type of the sentiment is not available in the manual coding data.

### 2.4. Coding Procedure

The textual feature coding scheme was built on previous literature and took into consideration the characteristics of tweets. The coding scheme broadly separates textual features into the semantic and word levels, each of which entails specific textual features (see Table 1).

Two coders, who were different from the ones involved in the manual coding for sentiment classification, were recruited. For each textual feature given a tweet, the coders were instructed to assign 1 when the textual feature was present, otherwise 0. The two coders coded 50 randomly sampled tweets and resolved inconsistencies through discussion in each round of the training sessions. After three rounds, an adequate level of mutual agreement was reached (Kripendoff’s alpha = 0.80). Two thousand tweets were randomly sampled and used for the main coding task. Thirty-one tweets were removed from the sample as they contained no text. The two coders coded the same 100 tweets out of the 1969 tweets to check the inter-coder reliability. The inter-coder reliability that the two coders reached was 0.86 in the main task. Confirming the substantial level of inter-coder reliability, each coder coded the rest of the tweets separately. Inconsistencies remaining in the 100 tweets used for the inter-coder reliability check were also resolved through discussion, and the 100 tweets were included in the main analysis.

### 2.5. Analytical Plan

The analysis consists of two parts. First, the validity of DSA was evaluated by comparing the sentiment classification of each application with that of the manual coding. Furthermore, the classification behavior of DSA was compared with each other. Second, a set of binary logistic regressions was conducted to examine the influences of the 11 textual features listed in Table 1 on the inconsistency between each DSA and the manual coding. The unit of analysis was a tweet, and the 11 textual features served as the independent variables in the binary logistic regression.

## 3. Results

### 3.1. DSA Validity Evaluation

Table 2 presents the overall evaluation results, indicating that DSA was inconsistent with the manual coding in terms of accuracy and macro-averaging F1 score. On average, the accuracy between the DSA and the manual coding was 33.53%. Specifically, LIWC showed the highest accuracy (56.84%), followed by orgSWN (37.46%), ANEW (32.87%), adSWN (21.25%), and SWN (19.22%).

When it comes to the specific sentiments, the DSA generally performed better at classifying neutral sentiment (mean of F1 scores (MF1, hereafter) = 0.37) than negative (MF1 = 0.29) and positive sentiment (MF1 = 0.18). The performances of DSA varied across the type of sentiments. SWN performed slightly better to identify negative sentiment (F1 = 0.35) than LIWC (F1 = 0.34), adSWN (F1 = 0.31), ANEW (F1 = 0.20), and orgSWN (F1 = 0.24). LIWC outperformed to discern neutral sentiment (F1 = 0.70) compared to orgSWN (F1 = 0.51), ANEW (F1 = 0.47), adSWN (F1 = 0.17), and SWN (F1 = 0.01). LIWC also performed better to classify positive sentiment (F1 = 0.30) than SWN (F1 = 0.19), orgSWN (F1 = 0.15), ANEW (F1 = 0.13), and adSWN (F1 = 0.13).

The study further examined to what extent the sentiment classification of the five DSAs are comparable with each other. Table 3 presents the results. In sum, LIWC’s classification was the most comparable with that of other DSA (averaged matched cases = 41.72%), which means that LIWC’s negative, neutral, positive, and mixed sentiment outputs are more likely to overlap with other DSA on average (i.e., convergent validity). On the other hand, adSWN’s classification output was the least comparable with that of other DSA (averaged matched cases = 30.99%).

### 3.2. Textual Features Associated with Invalidity of DSA

Table 4 presents the overall results of the binary logistic regressions. The findings showed that the presence of certain textual features made certain DSA invalid compared to the manual coding approach. For the semantic level features, the presence of irrealis (b = 0.47, *p* < 0.05) and negations (b = 0.77, *p* < 0.001) made LIWC invalid. Furthermore, the presence of negations (b = 0.42, *p* < 0.01) and intensifiers (b = 0.35, *p* < 0.05) were likely to make orgSWN produce invalid sentiment classifications. For the word level features, the presence of abbreviations (b = 0.46, *p* < 0.05) and acronyms (b = 1.00, *p* < 0.05) significantly contributed to the likelihood of misclassifying sentiment in LIWC. The presence of irregularly capitalized words (b = 0.83, *p* < 0.05) and abbreviations (b = 0.54, *p* < 0.05) increased the likelihood of misclassification in ANEW. The presence of acronyms (b = 1.26, *p* < 0.05) increased the likelihood of misclassification in orgSWN.

Unexpectedly, for SWN and adSWN, the likelihood of misclassification was reduced upon the presence of textual features. For instance, the presence of irrealis reduced the likelihood of misclassification in SWN (b = −1.06, *p* < 0.001), and the presence of embedded hashtags decreased the likelihood of misclassification in adSWN (b = −0.29, *p* < 0.05). The presence of negations reduced the likelihood of misclassification in both SWN (b = −1.17, *p* < 0.001) and adSWN (b = −0.60, *p* < 0.001). Other textual features such as sarcasm, unconfirmed typos, irregularly capitalized words, abbreviations, and acronyms reduced the likelihood of misclassification of SWN and/or adSWN (see Table 4 for the detail). The unexpected findings will be discussed further in the discussion section.

## 4. Discussion

DSA has been widely used in the field of public health research. However, our findings and previous literature suggest that scholars should utilize DSA with caution [10]. Our empirical evaluation indicates that DSA widely used for public health research may not be completely accurate, at least in the context of infectious disease in identifying sentiments when the manual coding approach served as the ground truth. Moreover, the sentiment classification behaviors diverge across DSA, which may indicate the low convergent validity of DSA. Failure to incorporate relevant textual features into DSA partially explains the discrepancies in sentiments identified between DSA and human coding.

DSA’s inability to account for semantic textual features such as irrealis, negations, and intensifiers and word-level features such as irregularly capitalized words, abbreviations, and acronyms, seems problematic at least for certain applications. Irrealis can change the meaning of sentiment-bearing words in a subtle manner [11]. The nuanced change in sentiment would make computational sentiment analysis, especially DSA, challenging to detect correct sentiment compared to the manual coding approach. Given that irrealis would be more frequently used in an uncertain situation such as a public health crisis, failure to account for irrealis might be problematic in public health research. Negations and intensifiers as valence shifters directly influence the sentiments of tweets. Negations play a role in reversing sentiments from being positive to negative or negative to positive. Intensifiers also change the sentiment of a tweet by magnifying the degree of the expressed sentiment. Given that negations and intensifiers are quite commonly used in our daily life and on social media and considerably shift the direction or the strength of sentiment, failure to account for these semantic level textual features may lead DSA to misclassify sentiments. Word-level textual features, including irregularly capitalized words, abbreviations, and acronyms, are also quite commonly used and can denote substantial sentiments in text-based user-generated content. Ignoring those features may lead to the invalidity of DSA.

It is noteworthy that not all the unrecognizable textual features significantly exacerbate the validity of DSA. Textual features such as sarcasm, diminishers, and lengthened words that did not significantly impact the validity of DSA, may be weakly or not associated with the sentiment of texts in the Ebola context. However, the findings should be interpreted with caution. Our data contained few cases for these textual features (i.e., sarcasm, diminishers, and lengthened words) as noted in Table 4. The small cases of textual features might hinder finding their significant influences on the validity of DSA. Future studies are needed to investigate the relationships between these textual features and the validity of DSA with more data for a robust conclusion.

The findings with SWN and adSWN indicating that the presence of some textual features improves the validity of DSA are unexpected. Although it is unclear why such unexpected results were found, it is speculated that the results could be biased due to the imbalanced distribution of the dependent variable. With low accuracy, SWN and adSWN included many more inconsistent cases than consistent cases (about four times more). Given that imbalanced data can lead to biased parameter estimations [27], the estimates of SWN and adSWN could be biased. Future studies are solicited to closely examine heterogeneous influences of textual features on misclassification among different DSA while accounting for such imbalanced data distribution.

We should acknowledge that textual features examined in the study are not comprehensive. Other textual features such as punctuations that may contribute to the invalidity of DSA should be examined in future research [28]. Moreover, text-based symbols such as emojis are also needed to be investigated in future research. Emojis are commonly embedded in online texts and have a significant influence on sentiments [29]. However, sentiments denoted in emojis are not generally capturable by DSA, which means that the presence of emojis can be associated with the invalidity of DSA. Future studies are solicited to replicate the current study with an extended list of textual features and symbols. Given that the performance of DSA can be gradually improved by addressing each of these unaccountable features, such replications will contribute to enhancing DSA.

### Suggestions for Scholars on the Use of DSA

Overall, the study helps scholars be mindful and aware of potential pitfalls when using DSA for public health research (especially for infectious disease-related research). Three suggestions can be outlined based on the findings: (1) scholars should be careful and critical about findings in studies that use DSA; (2) certain textual features should be more carefully addressed in DSA; and (3) alternatively, scholars can adopt advanced DSA (e.g., VADER, SO-CAL) that can deal with the problematic textual features identified from the study or machine learning approach (e.g., support vector machine).

Scholars should be critical about health research using DSA without validation. Here, we do not try to discredit the whole body of scholarly public health research utilizing DSA. This single study has its limitations and does not represent a comprehensive evaluation of DSA. Rather, this study points out that scholars should be aware of the potential methodological menace of DSA. Given that using an invalid measurement will lead researchers to reach the wrong conclusion, which is very critical and dangerous in science [30], a critical evaluation of public health research using DSA is necessary. Furthermore, we strongly suggest that the DSA’s validity evaluation should be mandatory in public health research. In general, the supervised machine learning approach seems more reliable. This is not because the approach is simply better than the DSA approach. Rather it is because most research using the supervised machine learning approach is accompanied with validation by default [31]. The fact that validation of DSA has been evaluated in other previous studies does not guarantee its validity is preserved in their current studies. Given that the performance of DSA is sensitive to many factors, such as text context and the pre-processing technique, it is difficult to justify that the pre-established validity of DSA is generalizable to other research. Therefore, it is necessary to have a critical view of findings derived from DSA, and it is highly desirable to make DSA validation mandatory by default in scientific journals.

The study suggests that certain semantic (i.e., irrealis, negations, intensifiers) and word-level features (i.e., irregularly capitalized words, abbreviations, and acronyms) should be more carefully addressed in DSA. Although these problematic textual features have been recognized by scholars [11], how the presence of the textual features actually deteriorates the validity of DSA has never been systematically and empirically investigated. Identifying textual features that considerably influence misclassification in DSA is a unique finding in the current study and provides practical implications for improving the DSA methods.

Alternatively, scholars can consider adopting a more advanced DSA (e.g., VADER and SO-CAL) that can deal with the problematic textual features identified from the study or using a machine learning approach. For instance, using VADER and SO-CAL that account for semantic textual features such as negations, intensifiers, and/or irrealis would help reduce the invalidity caused by those semantic textual features [11,32]. In addition, given that abbreviations and acronyms are commonly used in online text and can denote substantial levels of sentiments [33], VADER that includes these informal words in their sentiment dictionaries could improve the classification performances [32]. A post-hoc examination of the current study indicates that VADER and SO-CAL mitigated the impact of problematic textual features (See Appendix A). Textual features such as irrealis, negations, and acronyms associated with the invalidity of the tested DSA were no longer significant when using VADER or SO-CAL. However, it should be noted that resolving problematic textual features does not necessarily improve the validity of DSA, as sentiments of texts can be expressed in complex and numerous ways. Indeed, although VADER and SO-CAL resolve the impact of problematic textual features to some extent, still the validities of VADER and SO-CAL do not reach a satisfactory level (See Appendix A). The machine learning approach can be another alternative. In the post-hoc evaluation of the current study, a support vector machine substantially outperforms any DSA (See Appendix A). Previous studies also suggest that a machine learning approach generally shows a better performance than DSA [10]. However, machine learning is subjected to other validity issues such as overfitting. Moreover, the failure to obtain high-quality training data can easily lead to invalidity. Researchers should be informed about the nature of their data and characteristics of each sentiment analysis method when choosing sentiment analysis. More importantly, researchers should do their best to thoroughly evaluate the validity of their sentiment method in their studies.

## 5. Conclusions

DSA has been widely adopted for public health research. However, previous literature suggests that scholars should utilize DSA with caution due to the lack of validity compared to the manual coding method [10]. The current study evaluates the validity of DSA and examines potential textual features associated with the invalidity of DSA. Our empirical evaluation indicates that DSA commonly used for public health research may not be completely accurate at least in the context of infectious disease in identifying sentiments when the manual coding approach served as the ground truth. Failure to integrate semantic textual features such as irrealis, negations, and intensifiers as well as word-level features such as irregularly capitalized words, abbreviations, and acronyms, is associated with the invalidity of certain DSA. Based on the findings, the current study draws three suggestions for public health researchers: (1) scholars should be careful and critical about findings in studies that use DSA, (2) certain textual features should be more carefully addressed in DSA, and (3) alternatively, scholars can adopt advanced DSA and a machine learning approach that address those problematic text features.

## Figures and Tables

**Table 1 ijerph-19-06759-t001:** Explanations and examples of textual features.

Textual Features	Definitions	Reasonings and Examples	References
**Semantic Level**			
Embedded Hashtag	A hashtag that grammatically structures a sentence.	An embedded hashtag can threaten the validity of DSA because a hashtag structurally embedded in tweets can be meaningful, and widely used but cannot be generally captured by DSA. (e.g., #Stopspreadingebola by donating $5 to our NGO.)	N.A.
Irrealis	A function indicating that a certain situation or action is unknown to happen.	It is challenging to estimate the accurate sentiment of a text containing irrealis because irrealis can change the meaning of sentiment-bearing words in a subtle manner. Irrealis’ markers include modal verbs (e.g., would, could, would have), conditional markers (e.g., if), negative polarity items (e.g., any, anything), certain verbs (e.g., expect, doubt, assume), and questions. (e.g., if it spreads, it will destroy everything it touches.)	[11]
Sarcasm	A sarcastic statement is defined as one where the opposite meaning is intended.	Sarcasm completely shifts the orientation of sentiment by using the opposite meaning of words given a context. (e.g., What did I tell you? This may be the “great plague.”)	[12]
Negation	Negations are terms that reverse the sentiment of a certain word.	Negations change the orientation of a sentence from positive to negative or negative to positive (e.g., no, not, rather, never, none, nobody, no one, nothing, neither, nor, nowhere, without). (e.g., Ebola ain’t fun.)	[11,13,14,15]
Intensifier	Intensifiers are terms that intensify the degree of the expressed sentiment.	Intensifiers change the sentiment of a sentence by intensifying the strength of sentiment (e.g., very, really, extraordinarily, huge, total). (e.g., the risk of Ebola infection for travelers is very low.)	[11,14,15]
Diminisher	Diminishers are terms that decrease the degree of the expressed sentiment.	Diminishers change the sentiment of a sentence by decreasing the strength of sentiment (e.g., slightly; somewhat; minor). (e.g., I’m a little worried about Ebola.)	[11,14]
**Word-Level**			
Unconfirmed typo	A misspelled word.	A misspelled word may hold sentiments but is not generally capturable by DSA. (e.g., I feel bad about Ebola.)	[16]
Lengthened word	A lengthened word.	A lengthened word is difficult to be captured through DSA due to its unstructured format, although it may contain stronger sentiment compared with an ordinary format word. (e.g., who’s got the biggest smile to save N lives against Ebola? Nooooobody.)	[16,17]
Irregularly capitalized word	A word that is capitalized in an uncommon way.	An irregularly capitalized word may contain stronger sentiment than a word in its ordinary format but is not generally capturable by DSA. (e.g., I think Bill Gates is a GREAT man!)	[18]
Abbreviation	A shortened form of a word.	An abbreviation may contain a sentiment but is generally ignored by DSA. (e.g., there is no cure or something is really bs.)	[16]
Acronym	A shortened form of a phrase that consists of the initials of each word.	An acronym may contain a sentiment but is generally ignored by DSA. (e.g., who TF eats bats?)	[19]

**Table 2 ijerph-19-06759-t002:** Validity evaluation in comparison with manual coding results.

	LIWC	ANEW	SWN	orgSWN	adSWN	
	F1	F1	F1	F1	F1	Mean
Neg	0.34	0.20	0.35	0.24	0.31	0.29
Neu	0.70	0.47	0.01	0.51	0.17	0.37
Pos	0.30	0.13	0.19	0.15	0.13	0.18
Macro Average	0.45	0.27	0.18	0.30	0.21	0.28
Accuracy (%)	56.84	32.87	19.22	37.46	21.25	33.53
Tweets (*n*)	7421	7797	7790	7319	7175	7500

Note: The number of tweets differs among applications due to the exclusion of tweets classified as mixed sentiment.

**Table 3 ijerph-19-06759-t003:** Sentiment classification comparison among DSA.

		LIWC				ANEW				SWN				orgSWN				Averaged Matched Cases
		Neg	Neu	Pos	Mix	Neg	Neu	Pos	Mix	Neg	Neu	Pos	Mix	Neg	Neu	Pos	Mix	
LIWC	Neg																	41.72
	Neu																	
	Pos																	
	Mix																	
ANEW	Neg	11.22	8.69	1.54	1.38													34.40
	Neu	6.80	21.67	6.10	1.24													
	Pos	8.62	20.66	9.85	2.21													
	Mix	0.00	0.01	0.00	0.01													
SWN	Neg	18.58	36.02	11.45	2.99	16.16	24.89	27.97	0.03									33.72
	Neu	0.08	0.22	0.08	0.01	0.08	0.17	0.14	0.00									
	Pos	7.98	14.71	5.95	1.83	6.59	10.71	13.17	0.00									
	Mix	0.00	0.09	0.01	0.01	0.01	0.05	0.05	0.00									
orgSWN	Neg	10.68	15.28	3.05	1.40	10.62	8.48	11.31	0.01	21.37	0.06	8.96	0.01					31.36
	Neu	9.08	20.82	4.62	1.26	6.04	16.25	13.48	0.01	24.73	0.19	10.76	0.09					
	Pos	5.28	11.92	8.51	1.94	4.80	8.83	14.03	0.00	18.78	0.12	8.74	0.01					
	Mix	1.59	3.00	1.31	0.26	1.38	2.26	2.51	0.00	4.14	0.01	2.00	0.00					
adSWN	Neg	13.51	22.49	6.82	2.33	11.19	16.98	16.99	0.00	31.18	0.18	13.73	0.06	21.37	0.06	8.96	0.01	30.99
	Neu	1.56	6.14	0.91	0.21	1.77	4.50	2.55	0.00	6.23	0.08	2.50	0.01	24.73	0.19	10.76	0.09	
	Pos	9.51	18.02	8.57	1.92	8.00	11.67	18.32	0.03	26.18	0.13	11.68	0.03	18.78	0.12	8.74	0.01	
	Mix	2.04	4.39	1.19	0.38	1.87	2.67	3.46	0.00	5.44	0.00	2.55	0.01	4.14	0.01	2.00	0.00	
Average of Total Matched Cases																		34.44

Note: Values are expressed in percentage, and values in a diagonal of a matrix between a pair of DSAs represent the percentage of consistent sentiment classification cases. For instance, both LIWC and ANEW classified 11.22% of tweets as negative, 21.67% of tweets as neutral, 9.85% of tweets as positive, and 0.01% as mixed sentiments. The sum of these values in the diagonal indicates the proportion of matched cases between LIWC and ANEW (42.75%). Averaging the proportion of matched cases between LIWC and other DSA is represented as averaged matched cases (41.72%).

**Table 4 ijerph-19-06759-t004:** The results of binary logistic regression: influences of textual features on inconsistency.

Textual Features (IVs)	Inconsistency (DV)				
	LIWC	ANEW	SWN	orgSWN	adSWN
Intercept	−0.42 ***	0.73 ***	1.83 ***	0.49 ***	1.66 ***
**Semantic Level**					
Embedded hashtags	−0.08	−0.16	−0.08	0.13	−0.29 *
Irrealis	0.47 *	0.21	−1.06 ***	−0.11	−0.37
Sarcasm	1.09	0.94	−0.32	−0.34	−1.69 *
Negations	0.77 ***	0.09	−1.17 ***	0.42 **	−0.60 ***
Intensifiers	0.21	−0.08	−0.30	0.35 *	−0.28
Diminishers	0.33	−0.25	−0.23	0.85	−0.33
**Word-level**					
Unconfirmed typos	0.62	0.30	−1.55 ***	0.24	−0.95 **
Lengthened words	0.95	0.93	−0.72	0.13	0.69
Irregularly capitalized words	0.45	0.83 *	−1.08 ***	0.29	−0.60 *
Abbreviations	0.46 *	0.54 *	−0.78 **	0.19	−0.38
Acronyms	1.00 *	0.52	−1.14 **	1.26 *	−0.78

Note: * *p* < 0.05, ** *p* < 0.01, *** *p* < 0.001; in DV, consistent condition = 0, inconsistent condition = 1; the number of tweets that include each of the textual features are as follows: embedded hashtags (*n* = 429), irrealis (*n* = 429), sarcasm (*n* = 7), negation (*n* = 248), intensifiers (*n* = 260), diminishers (*n* = 11), unconfirmed typos (*n* = 35), lengthened words (*n* = 7), irregularly capitalized words (*n* = 71), abbreviations (*n* = 88), acronyms (*n* = 30); the total sample size is 1969.

## Data Availability

Data are available from the corresponding author upon reasonable request.

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
