# Peer review of "Detecting Sentiment toward Emerging Infectious Diseases on Social Media: A Validity Evaluation of Dictionary-Based Sentiment Analysis"

_ijerph, 2022, doi:10.3390/ijerph19116759_

Round 1
Reviewer 1 Report
Review for
Detecting Sentiment toward Emerging Infectious Diseases on 2 Social Media: A Validity Evaluation of Dictionary-based Sentiment Analysis
Sanguk Lee, Siyuan Ma, Jingbo Meng, Jie Zhuang, Tai-Quan Peng
In this article, the authors make the case that public health researchers who utilize computational sentiment analysis on digital text from social media posts should be cautious when relying solely on dictionary-sentiment analysis (DSA) tools that do not account for important text-based indicators of expressed sentiment. The authors describe various semantic-level features (such as irrealis, negations, intensifiers, diminishers, sarcasm, embedded hashtags) and word-level features (such as irregularly capitalized words, sentiment-laden abbreviations and acronyms, typos or intentionally lengthened/exaggerated spellings). The study presents the F1-score accuracy for five DSA tools (LIWC, ANEW, SWN, orgSWN, and adSWN) by comparing them to the ratings of two human judges. The study also assessed the degree to which each textual feature contributed to the invalidity/inaccurate sentiment classifications by the DSA tools.
Based on the empirical evidence provided in Table 2, the authors conclude that simplistic DSA techniques lack validity and that public health scholars should utilize DSA with caution and be critical about findings in public health research that uses DSA. The authors also advocate that applications of DSA in public health research should more carefully address the text-based indicators and sentiment analysis features discussed, or adopt more advanced dictionary-based techniques such as SO-CAL or VADER, or else rely on ML approaches (which the authors claim are not necessarily better but do tend to report validation results more often).
In terms of novelty, the issues related to simplistic bag-of-words / dictionary techniques have been known for almost two decades, and popular well-established solutions have been developed (e.g., VADER) that have also been around for nearly ten years now. Therefore, while I do not really consider the authors conclusion to be cautious/critical of outdated techniques to be a significant contribution to the field of text-based analysis of sentiment, what I did find useful in this paper was the presentation of specifically which features were the biggest (and sometimes unexpected) correlates to invalidity in simplistic DSA methods/tools. To me, the discussion in section 3.2 and data in Table 3 represent the material substance providing advancement of the current knowledge (that is, unless further changes are made to the paper, see below for suggestions).
In terms of scope, I believe the topic and content of the paper are a good fit for the International Journal of Environmental Research and Public Health. The quality of writing is good.
In terms of significance of findings and justification of conclusions, it seems there is very little evidence in the manuscript to support the advocacy of SO-CAL, VADER, or other ML approaches. This minor shortfall could easily be addressed by integrating (e.g., directly into Table 2) a side-by-side comparison of the accuracy of sentiment analysis results from advanced approaches such as VADER, SO-CAL, and select ML models right alongside the simple DSA techniques already presented.
Reviewer 2 Report
Dear author,
The manuscript title is “Detecting sentiment toward emerging infectious diseases on social media: a validity evaluation of dictionary-based sentiment analysis” and it aims to evaluate the validity of DSA applications compared to the manual coding approach in the case of Ebola.
The topic falls within the aims and scope of the journal.
I did not find any improvement needed as it is seemed to me very clear and well done. If mandatory to write something here, maybe I would add more human manual coders as they used two each time.
I have nothing to add in addition to formatting issues that I believe the journal warns about, namely the fact that when the table is broken into two pages, the title must be repeated and added (cont.).
Although not totally “new” information is given, the study is very interesting and useful I believe.
Reviewer 3 Report
Dear Authors
Thank you very much for conducting this study.
I suggest that it is reconsider after major revision as suggested below.
Minor concerns:
1. DSA is defined as "dictionary-based sentiment analysis (DSA)". Throughout the text the expression "DSA applications" should be replaced by "DSA" only.
Major concerns:
1. It is well accepted in the literature that the validity of sentiment analysis highly depends on the domain. The authors stated in the title that they are evaluation DSA validity for emerging infectious diseases and Ebola tweets have been used for data analysis. Nevertheless the authors incorrectly try to generalise their findings to public health research in general. Conclusions like "Our empirical evaluation indicates that DSA applications commonly used for public health research may not be completely accurate in identifying sentiments [...]" and "Based on the findings, the current study draws three suggestions for public health researchers [...]" and similar statements throughout the text, lack of scientific soundness and can not be claimed based in this study. From this study suggestions might be drawn for Ebola researchers, eventually for emerging infectious diseases researchers if well justified, but not for public health researchers in general. The authors must re-write their conclusions or conduct another study that allows more general conclusions.
2. Authors don't address the emojis issue as a textual feature. Given its generalised use, not addressing this issue should be justified.
3. Intra-coder reliability is missing.
4. Relabelled fear as negative might be dubious in public health DSA.
5. Regarding "Textual Features Causing Invalidity of DSA", no causality analysis have been conducted.
6. Some textual features require more data to be included in any analysis: sarcasm (n = 7), diminishers (n = 11), lengthened words (n = 7)
Reviewer 4 Report
the work is of interest, the style is correct and the results are clearly presented. However, the results in table 3 suggest to me that the outputs of the different applications are quite divergent. These results set precedents for artificial intelligence specialists to work more on these aspects. I would find it interesting if the authors add a table comparing the behavior of the 5 applications with each other.
I also suggest making some corrections:
33 ... is the off-the-shelf Dictionary-based Sentiment Analysis (DSA, hereafter), which estimates sentiments by simply retrieving sentiment scores of words from a predefined sentiment dictionary.
(add references)
164 When it comes to the specific sentiments, the DSA applications generally performed better at classifying neutral sentiment (MF1 = .37) than negative (MF1 = .29) and positive ... MF1???
250 3) alternatively, scholars can adopt advanced DSA applications (e.g., SO-CAL; VADER).
what other differences exist between these applications and those used at work? Why weren't they evaluated?
259 . In general, the supervised machine learning approach seems more reliable and transparent (add reference/s)
Round 2
Reviewer 3 Report
Dear Authors,
Thank you for addressing my concerns. I'm happy with your comments and changes in the manuscript.
Kind regards.